# Incorporating Interpretable Output Constraints in Bayesian Neural Networks

**Wanqian Yang**
Harvard University
Cambridge, MA, USA
yangw@college.harvard.edu

**Lars Lorch**[*]
ETH Zürich
8092 Zürich, Switzerland
llorch@student.ethz.ch

**Moritz A. Graule**
Harvard University
Cambridge, MA, USA
graulem@g.harvard.edu

**Himabindu Lakkaraju**
Harvard University
Cambridge, MA, USA
hlakkaraju@seas.harvard.edu

**Finale Doshi-Velez**
Harvard University
Cambridge, MA, USA
finale@seas.harvard.edu

## Abstract

Domains where supervised models are deployed often come with task-specific constraints, such as prior expert knowledge on the ground-truth function, or desiderata like safety and fairness. We introduce a novel probabilistic framework for reasoning with such constraints and formulate a prior that enables us to effectively incorporate them into Bayesian neural networks (BNNs), including a variant that can be amortized over tasks. The resulting Output-Constrained BNN (OC-BNN) is fully consistent with the Bayesian framework for uncertainty quantification and is amenable to black-box inference. Unlike typical BNN inference in uninterpretable parameter space, OC-BNNs widen the range of functional knowledge that can be incorporated, especially for model users without expertise in machine learning. We demonstrate the efficacy of OC-BNNs on real-world datasets, spanning multiple domains such as healthcare, criminal justice, and credit scoring.

## 1 Introduction

In domains where predictive errors are prohibitively costly, we desire models that can both capture predictive uncertainty (to inform downstream decision-making) as well as enforce prior human expertise or knowledge (to induce appropriate model biases). Performing Bayesian inference on deep neural networks, which are universal approximators [11] with substantial model capacity, results in BNNs — models that combine high representation power with quantifiable uncertainty estimates [21, 20] [1]. The ability to encode informative functional beliefs in BNN priors can significantly reduce the bias and uncertainty of the posterior predictive, especially in regions of input space sparsely covered by training data [27]. Unfortunately, the trade-off for their versatility is that BNN priors, defined in high-dimensional parameter space, are uninterpretable. A general approach for incorporating functional knowledge (that human experts might possess) is therefore intractable.

Recent work has addressed the challenge of incorporating richer functional knowledge into BNNs, such as preventing miscalibrated model predictions out-of-distribution [9], enforcing smoothness constraints [2] or specifying priors induced by covariance structures in the dataset (cf. Gaussian processes) [25, 19]. In this paper [2], we take a different direction by tackling functional knowledge expressed as *output constraints* — the set of values $y$ is constrained to hold for any given $\mathbf{x}$. Unlike other types of functional beliefs, output constraints are intuitive, interpretable and easily specified,

---

[*]Work done while at Harvard University.

[1]See Appendix A for a technical overview of BNN inference and acronyms used throughout this paper.

[2]Our code is publicly available at: https://github.com/dtak/ocbnn-public.

even by domain experts without technical understanding of machine learning methods. Examples include ground-truth human expertise (e.g. known input-output relationship, expressed as scientific formulae or clinical rules) or critical desiderata that the model should enforce (e.g. output should be restricted to the permissible set of safe or fair actions for any given input scenario).

We propose a sampling-based prior that assigns probability mass to BNN parameters based on how well the BNN output obeys constraints on drawn samples. The resulting **Output-Constrained BNN** (OC-BNN) allows the user to specify any constraint directly in its functional form, and is amenable to all black-box BNN inference algorithms since the prior is ultimately evaluated in parameter space.

Our contributions are: **(a)** we present a formal framework that lays out what it means to learn from output constraints in the probabilistic setting that BNNs operate in, **(b)** we formulate a prior that enforces output constraint satisfaction on the resulting posterior predictive, including a variant that can be amortized across multiple tasks, **(c)** we demonstrate proof-of-concepts on toy simulations and apply OC-BNNs to three real-world, high-dimensional datasets: (i) enforcing physiologically feasible interventions on a clinical action prediction task, (ii) enforcing a racial fairness constraint on a recidivism prediction task where the training data is biased, and (iii) enforcing recourse on a credit scoring task where a subpopulation is poorly represented by data.

## 2 Related Work

**Noise Contrastive Priors**  Hafner et al. [9] propose a generative "data prior" in function space, modeled as zero-mean Gaussians if the input is out-of-distribution. Noise contrastive priors are similar to OC-BNNs as both methods involve placing a prior on function space but performing inference in parameter space. However, OC-BNNs model output constraints, which encode a richer class of functional beliefs than the simpler Gaussian assumptions encoded by NCPs.

**Global functional properties**  Previous work have enforced various functional properties such as Lipschitz smoothness [2] or monotonicity [29]. The constraints that they consider are different from output constraints, which can be defined for local regions in the input space. Furthermore, these works focus on classical NNs rather than BNNs.

**Tractable approximations of stochastic process inference**  Garnelo et al. [7] introduce neural processes (NP), where NNs are trained on sequences of input-output tuples $\{(\mathbf{x}, y)_i\}_{i=1}^m$ to learn distributions over functions. Louizos et al. [19] introduce a NP variant that models the correlation structure of inputs as dependency graphs. Sun et al. [25] define the BNN variational objective directly over stochastic processes. Compared to OC-BNNs, these models represent a distinct direction of work, since (i) VI is carried out directly over function-space terms, and (ii) the set of prior functional assumptions is different; they cannot encode output constraints of the form that OC-BNNs consider.

**Equality/Inequality constraints for deep probabilistic models**  Closest to our work is that of Lorenzi and Filippone [18], which incorporates equality and inequality constraints, specified as differential equations, into regression tasks. Similar to OC-BNNs, constraint ($\mathcal{C}$) satisfaction is modeled as the conditional $p(Y, \mathcal{C}|\mathbf{x})$, e.g. Gaussian or logistic. However, we (i) consider a broader framework for reasoning with constraints, allowing for diverse constraint formulations and defined for both regression *and classification* tasks, and (ii) verify the tractability and accuracy of our approach on a comprehensive suite of experiments, in particular, high-dimensional tasks.

## 3 Notation

Let $\mathbf{X} \in \mathcal{X}$ where $\mathcal{X} = \mathbb{R}^Q$ be the input variable and $Y \in \mathcal{Y}$ be the output variable. For regression tasks, $\mathcal{Y} = \mathbb{R}$. For $K$-classification tasks, $\mathcal{Y}$ is any set with a bijection to $\{0, 1, \ldots, K-1\}$. We denote the ground-truth mapping (if it exists) as $f^* : \mathcal{X} \to \mathcal{Y}$. While $f^*$ is unknown, we have access to observed data $D_{tr} = \{\mathbf{x}_i, y_i\}_{i=1}^N$, which may be noisy or biased. A conventional BNN is denoted as $\Phi_{\mathbf{W}} : \mathcal{X} \to \mathcal{Y}$. $\mathbf{W} \in \mathcal{W}$, where $\mathcal{W} = \mathbb{R}^M$, are the BNN parameters (weights and biases of all layers) represented as a flattened vector. All BNNs that we consider are multilayer perceptrons with RBF activations. Uppercase notation denotes random variables; lowercase notation denotes instances.

## 4 Output-Constrained Priors

In joint $\mathcal{X} \times \mathcal{Y}$ space, our goal is to constrain the output $y$ for any set of inputs $\mathbf{x}$. In this setting, classical notions of "equality" and "inequality" (in constrained optimization) respectively become *positive* constraints and *negative* constraints, specifying what values $y$ can or cannot take.

**Definition 4.1.** *A **deterministic output constraint** $\mathcal{C}$ is a tuple $(\mathcal{C}_{\mathbf{x}}, \mathcal{C}_y, \circ)$ where $\mathcal{C}_{\mathbf{x}} \subseteq \mathcal{X}$, $\mathcal{C}_y : \mathcal{C}_{\mathbf{x}} \to 2^{\mathcal{Y}}$ and $\circ \in \{\in, \notin\}$. $\mathcal{C}$ is **satisfied** by an output $y$ iff $\forall \mathbf{x} \in \mathcal{C}_{\mathbf{x}}$, $y \circ \mathcal{C}_y(\mathbf{x})$. $\mathcal{C}^+ := \mathcal{C}$ is a **positive constraint** if $\circ$ is $\in$. $\mathcal{C}^- := \mathcal{C}$ is a **negative constraint** if $\circ$ is $\notin$. $\mathcal{C}$ is a **global constraint** if $\mathcal{C}_{\mathbf{x}} = \mathcal{X}$. $\mathcal{C}$ is a **local constraint** if $\mathcal{C}_{\mathbf{x}} \subset \mathcal{X}$.*

The distinction between positive and negative constraints is not trivial because the user typically has access to only one of the two forms. Positive constraints also tend to be more informative than negative constraints. Definition 4.1 alone is not sufficient as we must define what it means for a BNN, which learns a predictive *distribution*, to satisfy a constraint $\mathcal{C}$. A natural approach is to evaluate the probability mass of the prior predictive that satisfies a constraint.

**Definition 4.2.** *A deterministic output constraint $\mathcal{C} = (\mathcal{C}_{\mathbf{x}}, \mathcal{C}_y, \circ)$ is $\epsilon$-**satisfied** by a BNN with prior $\mathbf{W} \sim p(\mathbf{w})$ if, $\forall \mathbf{x} \in \mathcal{C}_{\mathbf{x}}$ and for some $\epsilon \in [0, 1]$:*

$$\int_{\mathcal{Y}} \mathbb{I}[y \circ \mathcal{C}_y(\mathbf{x})] \cdot p(\Phi_{\mathbf{W}} = y | \mathbf{x}) \, dy \quad \geq 1 - \epsilon \tag{1}$$

*where $p(\Phi_{\mathbf{W}} | \mathbf{x}) := \int_{\mathcal{W}} p(\Phi_{\mathbf{w}}(\mathbf{x}) | \mathbf{x}, \mathbf{w}) \, p(\mathbf{w}) \, d\mathbf{w}$ is the BNN prior predictive.*

The strictness parameter $\epsilon$ can be related to *hard* and *soft* constraints in optimization literature; the constraint is hard iff $\epsilon = 0$. Note that $\epsilon$ is not a parameter of our prior, instead, (1) is the goal of inference and can be empirically evaluated on a test set. Since BNNs are probabilistic, we can generalize Definition 4.1 further and specify a constraint directly as some distribution over $Y$:

**Definition 4.3.** *A **probabilistic output constraint** $\mathcal{C}$ is a tuple $(\mathcal{C}_{\mathbf{x}}, \mathcal{D}_y)$ where $\mathcal{C}_{\mathbf{x}} \subseteq \mathcal{X}$ and $\mathcal{D}_y(\mathbf{x})$ is a distribution over $\mathcal{Y}$. (That is, $\mathcal{D}_y$, like $\mathcal{C}_y$, is a partial function well-defined on $\mathcal{C}_{\mathbf{x}}$.) $\mathcal{C}$ is $\epsilon$-**satisfied** by a BNN with prior $\mathbf{W} \sim p(\mathbf{w})$ if, $\forall \mathbf{x} \in \mathcal{C}_{\mathbf{x}}$ and for some $\epsilon \in [0, 1]$:*

$$D_{DIV}\Big(p(\Phi_{\mathbf{W}} | \mathbf{x}) \, \big|\big| \, \mathcal{D}_y(\mathbf{x})\Big) \quad \leq \epsilon \tag{2}$$

*where $D_{DIV}$ is any valid measure of divergence between two distributions over $\mathcal{Y}$.*

We seek to construct a prior $\mathbf{W} \sim p(\mathbf{w})$ such that the BNN $\epsilon$-satisfies, for some small $\epsilon$, a specified constraint $\mathcal{C}$. An intuitive way to connect a prior in parameter space to $\mathcal{C}$ (defined in *function* space) is to evaluate how well the implicit distribution of $\Phi_{\mathbf{W}}$, induced by the distribution of $\mathbf{W}$, satisfies $\mathcal{C}$. In Section 4.1, we construct a prior that is conditioned on $\mathcal{C}$ and explicitly factors in the likelihood $p(\mathcal{C}|\mathbf{w})$ of constraint satisfaction. In Section 4.3, we present an amortized variant by performing variational optimization on objectives (1) or (2) directly.

## 4.1 Conditional Output-Constrained Prior

A fully Bayesian approach requires a proper distribution that describes how well any $\mathbf{w} \in \mathcal{W}$ satisfies $\mathcal{C}$ by way of $\Phi_{\mathbf{w}}$. Informally, this distribution must be conditioned on *all* $\mathbf{x} \in \mathcal{C}_{\mathbf{x}}$ (though we sample finitely during inference) and is the "product" of how well $\Phi_{\mathbf{w}}(\mathbf{x})$ satisfies $\mathcal{C}$ for *each* $\mathbf{x} \in \mathcal{C}_{\mathbf{x}}$. This notion can be formalized by defining $\mathcal{C}$ as a stochastic process indexed on $\mathcal{C}_{\mathbf{x}}$.

Let $\mathbf{x} \in \mathcal{C}_{\mathbf{x}}$ be any single input. In measure-theoretic terms, the corresponding output $Y : \Omega \to \mathcal{Y}$ is defined on some probability space $(\Omega, \mathcal{F}, \mathbf{P}_{g, \mathbf{x}})$, where $\Omega$ and $\mathcal{F}$ are the typical sample space and $\sigma$-algebra, and $\mathbf{P}_{g, \mathbf{x}}$ is any valid probability measure corresponding to a distribution $p_g(\cdot | \mathbf{x})$ over $\mathcal{Y}$. Then $(\boldsymbol{\Omega}, \mathbf{F}, \mathbf{P}_g)$ is the joint probability space, where $\boldsymbol{\Omega} = \prod_{\mathbf{x} \in \mathcal{C}_{\mathbf{x}}} \Omega$, $\mathbf{F}$ is the product $\sigma$-algebra and $\mathbf{P}_g = \prod_{\mathbf{x} \in \mathcal{C}_{\mathbf{x}}} \mathbf{P}_{g, \mathbf{x}}$ is the product measure (a pushforward measure from $\mathbf{P}_{g, \mathbf{x}}$). Let $\mathcal{CP} : \boldsymbol{\Omega} \to \mathcal{Y}^{\mathcal{C}_{\mathbf{x}}}$ be the stochastic process indexed by $\mathcal{C}_{\mathbf{x}}$, where $\mathcal{Y}^{\mathcal{C}_{\mathbf{x}}}$ denotes the set of all measurable functions from $\mathcal{C}_{\mathbf{x}}$ into $\mathcal{Y}$. The law of the process $\mathcal{CP}$ is $p(S) = \mathbf{P}_g \circ \mathcal{CP}^{-1}(S)$ for all $S \in \mathcal{Y}^{S_{\mathbf{x}}}$. For any finite subset $\{Y^{(1)}, \ldots, Y^{(T)}\} \subseteq \mathcal{CP}$ indexed by $\{\mathbf{x}^{(1)}, \ldots, \mathbf{x}^{(T)}\} \subseteq \mathcal{C}_{\mathbf{x}}$,

$$p(Y^{(1)} = y^{(1)}, \ldots, Y^{(T)} = y^{(T)}) = \prod_{t=1}^{T} p_g(Y^{(t)} = y^{(t)} | \mathbf{x}^{(t)}) \tag{3}$$

$\mathcal{CP}$ is a valid stochastic process as it satisfies both (finite) exchangeability and consistency, which are sufficient conditions via the Kolmogorov Extension Theorem [23]. As the BNN output $\Phi_{\mathbf{w}}(\mathbf{x})$ can be evaluated for all $\mathbf{x} \in \mathcal{C}_{\mathbf{x}}$ and $\mathbf{w} \in \mathcal{W}$, $\mathcal{CP}$ allows us to formally describe how much $\Phi_{\mathbf{w}}$ satisfies $\mathcal{C}$, by determining the measure of the realization $\Phi_{\mathbf{w}}^{\mathcal{C}_{\mathbf{x}}} \in \mathcal{Y}^{\mathcal{C}_{\mathbf{x}}}$.

**Definition 4.4.** *Let $\mathcal{C}$ be any (deterministic or probabilistic) constraint and $\mathcal{CP}$ the the stochastic process defined on $\mathcal{Y}^{\mathcal{C}_\mathbf{x}}$ for some measure $\mathbf{P}_g$. The **conditional output-constrained prior (COCP)** on $\mathbf{W}$ is defined as*

$$p_\mathcal{C}(\mathbf{w}) = p_f(\mathbf{w})p(\Phi_\mathbf{w}^{\mathcal{C}_\mathbf{x}}) \tag{4}$$

*where $p_f(\mathbf{w})$ is any distribution on $\mathbf{W}$ that is independent of $\mathcal{C}$, and $\Phi_\mathbf{w}^{\mathcal{C}_\mathbf{x}} \in \mathcal{Y}^{\mathcal{C}_\mathbf{x}}$ is the realization of $\mathcal{CP}$ corresponding to the BNN output $\Phi_\mathbf{w}(\mathbf{x})$ for all $\mathbf{x} \in \mathcal{C}_\mathbf{x}$.*

We can view (4) loosely as an application of Bayes' Rule, where the realization $\Phi_\mathbf{w}^{\mathcal{C}_\mathbf{x}}$ of the stochastic process $\mathcal{CP}$ is the evidence (likelihood) of $\mathcal{C}$ being satisfied by $\mathbf{w}$ and $p_f(\mathbf{w})$ is the unconditional prior over $\mathcal{W}$. This implies that the posterior that we are ultimately inferring is $\mathbf{W}|D_{tr}, \mathcal{CP}$, i.e. with extra conditioning on $\mathcal{CP}$. To ease the burden on notation, we will drop the explicit conditioning on $\mathcal{CP}$ and treat $p_\mathcal{C}(\mathbf{w})$ as the unconditional prior over $\mathcal{W}$. To make clear the presence of $\mathcal{C}$, we will denote the constrained posterior as $p_\mathcal{C}(\mathbf{w}|D_{tr})$. Definition 4.4 generalizes to multiple constraints $\mathcal{C}^{(1)}, \ldots, \mathcal{C}^{(m)}$ by conditioning on all $\mathcal{C}^{(i)}$ and evaluating $\Phi_\mathbf{w}$ for each constraint. Assuming the constraints to be mutually independent, $p_{\mathcal{C}^{(1)},\ldots,\mathcal{C}^{(m)}}(\mathbf{w}) = p_f(\mathbf{w})\prod_{i=1}^m p(\Phi_\mathbf{w}^{\mathcal{C}^{(i)}_\mathbf{x}})$.

It remains to describe what measure $\mathbf{P}_g$ we should choose such that $p(\Phi_\mathbf{w}^{\mathcal{C}_\mathbf{x}})$ is a likelihood distribution that faithfully evaluates the extent to which $\mathcal{C}$ is satisfied. For probabilistic output constraints, we simply set $p_g(\cdot|\mathbf{x})$ to be $\mathcal{D}_y(\mathbf{x})$. For deterministic output constraints, we propose a number of distributions over $\mathcal{Y}$ that corresponds well to $\mathcal{C}_y$, the set of permitted (or excluded) output values.

**Example: Positive Mixture of Gaussian Constraint** A positive constraint $\mathcal{C}^+$ for regression, where $\mathcal{C}_y(\mathbf{x}) = \{y_1, \ldots, y_K\}$ contains multiple values, corresponds to the situation wherein the expert knows potential ground-truth values over $\mathcal{C}_\mathbf{x}$. A natural choice is the Gaussian mixture model: $\mathcal{CP}(\mathbf{x}) \sim \sum_{k=1}^K \omega_k \mathcal{N}(y_k, \sigma_\mathcal{C}^2)$, where $\sigma_\mathcal{C}$ is the standard deviation of the Gaussian, a hyperparameter controlling the strictness of $\mathcal{C}$ satisfaction, and $\omega_k$ are the mixing weights: $\sum_{k=1}^K \omega_k = 1$.

**Example: Positive Dirichlet Constraint** For $K$-classification, the natural distribution to consider is the Dirichlet distribution, whose support (the standard $K$-simplex) corresponds to the $K$-tuple of predicted probabilities on all classes. For a positive constraint $\mathcal{C}^+$, we specify:

$$\mathcal{CP}(\mathbf{x}) \sim \mathcal{D}ir(\boldsymbol{\alpha}); \qquad \boldsymbol{\alpha}_i = \begin{cases} \gamma & \text{if } i \in \mathcal{C}_y(\mathbf{x}) \\ \gamma(1-c) & \text{otherwise} \end{cases} \qquad \text{where } \gamma \geq 1, \; 0 < c < 1 \tag{5}$$

**Example: Negative Exponential Constraint** A negative constraint $\mathcal{C}^-$ for regression takes $\mathcal{C}_y(\mathbf{x})$ to be the set of values that cannot be the output of $\mathbf{x}$. For the case where $\mathcal{C}_y$ is determined from a set of inequalities of the form $\{g_1(\mathbf{x}, y) \leq 0, \ldots, g_l(\mathbf{x}, y) \leq 0\}$, where *obeying all* inequalities implies that $y \in \mathcal{C}_y(\mathbf{x})$ and hence $\mathcal{C}^-$ is *not satisfied*, we can consider an exponential distribution that penalizes $y$ based on how each inequality is violated:

$$p_g(\mathcal{CP}(\mathbf{x}) = y'|\mathbf{x}) \propto \exp\left\{ -\gamma \cdot \prod_{i=1}^l \sigma_{\tau_0, \tau_1}(g_i(\mathbf{x}, y')) \right\} \tag{6}$$

where $\sigma_{\tau_0, \tau_1}(z) = \frac{1}{4}(\tanh(-\tau_0 z) + 1)(\tanh(-\tau_1 z) + 1)$ and $\gamma$ is a decay hyperparameter. The sigmoidal function $\sigma_{\tau_0, \tau_1}(g_i(\mathbf{x}, y'))$ is a soft indicator of whether $g_i \leq 0$ is satisfied. $p_g(\mathcal{CP}(\mathbf{x}) = y'|\mathbf{x})$ is small if every $g_i(\mathbf{x}, y') \leq 0$, i.e. all inequalities are obeyed and $\mathcal{C}^-$ is violated.

### 4.2 Inference with COCPs

A complete specification of (4) is sufficient for inference using COCPs, as the expression $p_\mathcal{C}(\mathbf{w})$ can simply be substituted for that of $p(\mathbf{w})$ in all black-box BNN inference algorithms. Since (4) cannot be computed exactly due to the intractability of $p(\Phi_\mathbf{w}^{\mathcal{C}_\mathbf{x}})$ for uncountable $\mathcal{C}_\mathbf{x}$, we will draw a finite sample $\{\mathbf{x}^{(t)}\}_{t=1}^T$ of $\mathbf{x} \in \mathcal{C}_\mathbf{x}$ (e.g. uniformly across $\mathcal{C}_\mathbf{x}$ or Brownian sampling if $\mathcal{C}_\mathbf{x}$ is unbounded) at the start of the inference process and compute (3) as an estimator of $p(\Phi_\mathbf{w}^{\mathcal{C}_\mathbf{x}})$ instead:

$$\tilde{p}_\mathcal{C}(\mathbf{w}) = p_f(\mathbf{w}) \prod_{t=1}^T p_g(\Phi_\mathbf{w}(\mathbf{x}^{(t)})|\mathbf{x}^{(t)}) \tag{7}$$

The computational runtime of COCPs increases with the input dimensionality $Q$ and the size of $\mathcal{C}_\mathbf{x}$. However, we note that many sampling techniques from statistical literature can be applied to COCPs in lieu of naive uniform sampling. We use the isotropic Gaussian prior for $p_f(\mathbf{w})$.

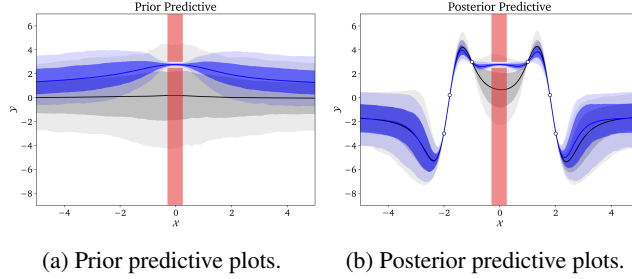

<center>(a) Prior predictive plots.      (b) Posterior predictive plots.</center>

Figure 1: 1D regression with the negative constraint: $\mathcal{C}_{\mathbf{x}}^{-} = [-0.3, 0.3]$ and $\mathcal{C}_y^{-} = (-\infty, 2.5] \cup [3, \infty)$, in red. The negative exponential COCP (6) is used. Even with such a restrictive constraint, the predictive uncertainty of the OC-BNN (blue) drops sharply to fit within the permitted region of $\mathcal{Y}$. Note that the OC-BNN posterior uncertainty matches the baseline (gray) everywhere except near $\mathcal{C}_{\mathbf{x}}$.

Absolute guarantees of (1) or (2), i.e. achieving $\epsilon = 0$, are necessary in certain applications but challenging for COCPs. Zero-variance or truncated $p_{\mathcal{C}}(\mathbf{w})$, as means of ensuring hard constraint satisfaction, are theoretically plausible but numerically unstable for BNN inference, particularly gradient-based methods. Nevertheless, as modern inference algorithms produce finite approximations of the true posterior, practical guarantees can be imposed, such as via further rejection sampling on top of the constrained posterior $p_{\mathcal{C}}(\mathbf{w}|D_{tr})$. We demonstrate in Section 5 that unlike naive rejection sampling from $p(\mathbf{w}|D_{tr})$, doing so from $p_{\mathcal{C}}(\mathbf{w}|D_{tr})$ is both tractable and practical for ensuring zero constraint violations.

### 4.3 An Amortized Output-Constrained Prior

Instead of constructing a PDF over $\mathbf{w}$ explicitly dependent on $\Phi_{\mathbf{w}}$ and $\mathcal{C}$, we can learn a variational approximation $q_{\boldsymbol{\lambda}}(\mathbf{w})$ where we optimize $\boldsymbol{\lambda}$ directly with respect to our goals, (1) or (2). As both objectives contain an intractable expectation over $\mathbf{W}$, we seek a closed-form approximation of the variational prior predictive $p_{\boldsymbol{\lambda}}(\Phi_{\mathbf{W}}|\mathbf{x})$. For the regression and *binary* classification settings, there are well-known approximations, which we state in Appendix B as (17) and (21) respectively. Our objectives are:

$$\boldsymbol{\lambda}^* = \arg\max_{\boldsymbol{\lambda} \in \Lambda} \int_{\mathcal{Y}} \mathbb{I}[y \circ \mathcal{C}_y(\mathbf{x})] \cdot p_{\boldsymbol{\lambda}}(\Phi_{\mathbf{W}} = y|\mathbf{x}) \, \mathrm{d}y \tag{8}$$

$$\boldsymbol{\lambda}^* = \arg\min_{\boldsymbol{\lambda} \in \Lambda} D_{DIV}\Big(p_{\boldsymbol{\lambda}}(\Phi_{\mathbf{W}}|\mathbf{x}) \,\big|\big|\, \mathcal{D}_y(\mathbf{x})\Big) \tag{9}$$

As (8) and (9) are defined for a specific $\mathbf{x} \in \mathcal{C}_{\mathbf{x}}$, we need to stochastically optimize over all $\mathbf{x} \in \mathcal{C}_{\mathbf{x}}$. Even though (8) is still an integral over $\mathcal{Y}$, it is tractable since we only need to compute the CDF corresponding to the boundary elements of $\mathcal{C}_y$. We denote the resulting learnt distribution $q_{\boldsymbol{\lambda}^*}(\mathbf{w})$ as the **amortized output-constrained prior (AOCP)**. Unlike COCPs, where $p_{\mathcal{C}}(\mathbf{w})$ is directly evaluated during posterior inference, we first perform optimization to learn $\boldsymbol{\lambda}^*$, which can then be used for inference independently over any number of training tasks (datasets) $D_{tr}$.

## 5 Low-Dimensional Simulations

COCPs and AOCPs are conceptually simple but work well, even on non-trivial output constraints. As a proof-of-concept, we simulate toy data and constraints for small input dimension and visualize the predictive distributions. See Appendix C for experimental details.

**OC-BNNs model uncertainty in a manner that respects constrained regions and explains training data, without making overconfident predictions outside $\mathcal{C}_{\mathbf{x}}$.** Figure 1 shows the prior and posterior predictive plots for a negative constraint for regression, where a highly restrictive constraint was intentionally chosen. Unlike the naive baseline BNN, the OC-BNN satisfies the constraint, with its predictive variance smoothly narrowing as $\mathbf{x}$ approaches $\mathcal{C}_{\mathbf{x}}$ so as to be entirely confined within $\mathcal{Y} - \mathcal{C}_y$. After posterior inference, the OC-BNN fits all data points in $D_{tr}$ closely while still respecting the constraint. Far from $\mathcal{C}_{\mathbf{x}}$, the OC-BNN is not overconfident, maintaining a wide variance like the baseline. Figure 2 shows an analogous example for classification. Similarly, the OC-BNN fits the data and respects the constraint within $\mathcal{C}_{\mathbf{x}}$, without showing a strong preference for any class OOD.

**OC-BNNs can capture global input-output relationships between $\mathcal{X}$ and $\mathcal{Y}$, subject to sampling efficacy.** Figure 3a shows an example where we enforce the constraint $xy \geq 0$. Even though the

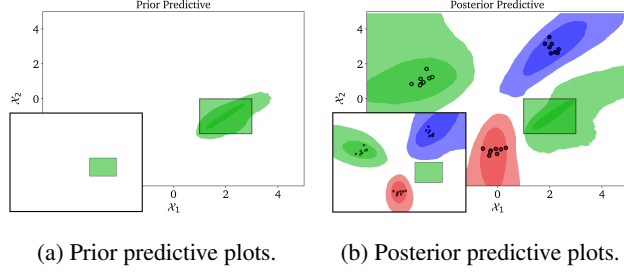

(a) Prior predictive plots.    (b) Posterior predictive plots.

Figure 2: 2D 3-classification with the positive constraint: $\mathcal{C}_{\mathbf{x}}^{+} = (1,3) \times (-2,0)$ and $\mathcal{C}_{y}^{+} = \{\text{green}\}$. The main plots are the OC-BNN predictives; insets are the baselines. Input region is shaded by the predicted class if above a threshold certainty. The positive Dirichlet COCP (5) is used. In both the prior and posterior, the constrained region (green rectangle) enforces the prediction of the green class.

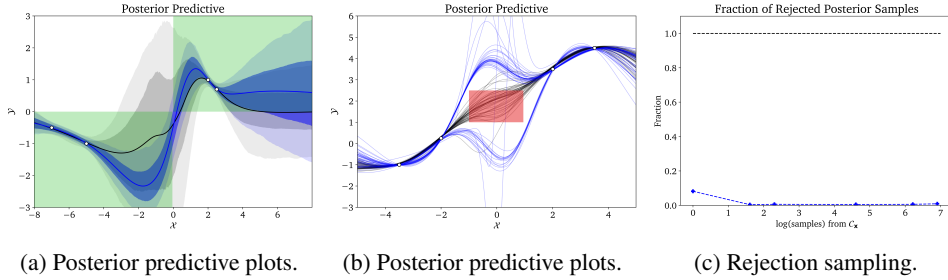

(a) Posterior predictive plots.    (b) Posterior predictive plots.    (c) Rejection sampling.

Figure 3: **(a)** 1D regression with the positive constraint: $\mathcal{C}_{\mathbf{x}}^{+} = \mathbb{R}$ and $\mathcal{C}_{y}^{+}(\mathbf{x}) = \{y \mid \mathbf{x} \cdot y \geq 0\}$ (green), using AOCP. **(b)** 1D regression with the negative constraint: $\mathcal{C}_{\mathbf{x}}^{-} = [-1,1]$ and $\mathcal{C}_{y}^{-} = [1, 2.5]$ (red), with the negative exponential COCP (6). The 50 SVGD particles represent functions passing above and below the constrained region, capturing two distinct predictive modes. **(c)** Fraction of rejected SVGD particles (out of 100) for the OC-BNN (blue, plotted as a function of log-samples used with COCP) and the baseline (black). All baseline particles were rejected, however, only 4% of particles were rejected, using just only 5 COCP samples.

training data itself adheres to this constraint, learning from $D_{tr}$ alone is insufficient. The OC-BNN posterior predictive narrows significantly (compared to the baseline) to fit the constraint, particularly near $x = 0$. Note, however, that OC-BNNs can only learn as well as sampling from $\mathcal{C}_{\mathbf{x}}$ permits.

**OC-BNNs can capture posterior multimodality.** As NNs are highly expressive, BNN posteriors can contain multiple modes of significance. In Figure 3b, the negative constraint is specified in such a way as to allow for functions that fit $D_{tr}$ to pass both above or below the constrained region. Accordingly, the resulting OC-BNN posterior predictive contains significant probability mass on either side of $\mathcal{C}_y$. Importantly, note that the negative exponential COCP does not *explicitly* indicate the presence of multiple modes, showing that OC-BNNs naturally facilitate mode exploration.

**OC-BNNs can model interpretable desiderata represented as output constraints.** OC-BNNs can be used to enforce important qualities that the system should possess. Figure 4 demonstrates a fairness constraint known as **demographic parity**:

$$p(Y = 1 | \mathbf{x}_A = 1) = p(Y = 1 | \mathbf{x}_A = 0) \tag{10}$$

where $\mathbf{x}_A$ is a protected attribute such as race or gender. (10) is expressed as a probabilistic output constraint. The OC-BNN not only learns to respect this constraint, it does so in the presence of *conflicting training data* ($D_{tr}$ is an unfair dataset).

**Sensitivity to Inference and Conflicting Data** Figure 5 in Appendix D shows the same negative constraint used in Figure 1, except using different inference algorithms. There is little difference in these predictive plots besides the idiosyncrasies unique to each algorithm. Hence **OC-BNNs behave reliably on, and are agnostic to, all major classes of BNN inference.** If $\mathcal{C}$ and $D_{tr}$ are incompatible, such as with adversarial or biased data, the posterior depends on (i) model capacity and (ii) factors affecting the prior and likelihood, e.g. volume of data or OC-BNN hyperparameters. For example, Figure 6 in Appendix D shows a BNN with enough capacity to fit both the noisy data

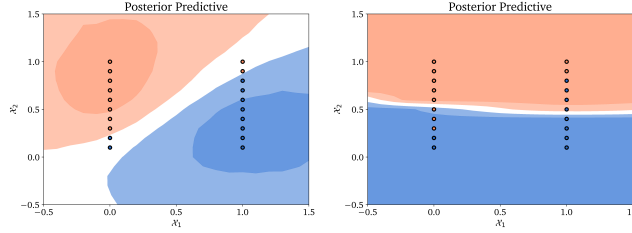

(a) Baseline posterior predictive.  (b) OC-BNN posterior predictive.

Figure 4: 2D binary classification. Suppose a hiring task where $\mathcal{X}_1$ (binary) indicates membership of a protected trait (e.g. gender or race) and $\mathcal{X}_2$ denotes skill level. Hence a positive (orange) classification should be correlated with higher values of $\mathcal{X}_2$. The dataset $D_{tr}$ displays historic bias, where members of the protected class ($\mathbf{x}_1 = 1$) are discriminated against ($Y = 1$ iff $\mathbf{x}_1 = 1, \mathbf{x}_2 \geq 0.8$, but $Y = 1$ iff $\mathbf{x}_1 = 0, \mathbf{x}_2 \geq 0.2$). A naive BNN **(a)** would learn an unfair linear separator. However, learning the probabilistic constraint: $\mathcal{D}_y(\mathbf{x})$ as the distribution where $p(\Phi(\mathbf{x}) = 1) = \mathbf{x}_2$ with the AOCP allows the OC-BNN **(b)** to learn a fair separator, **despite a biased dataset**.

as well as ground-truth constraints, resulting in "overfitting" of the posterior predictive. However, the earlier example in Figure 4 prioritizes the fairness constraint, as the small size of $D_{tr}$ constitutes weaker evidence.

**Ensuring Hard Constraints using Rejection Sampling**  A few SVGD particles in Figure 3b violate the constraint. While COCPs cannot guarantee $\epsilon = 0$ satisfaction, hard constraints can be *practically* enforced. Consider the idealized, truncated predictive distributions of the baseline BNN and OC-BNN, whereby for each $\mathbf{x} \in \mathcal{C}_\mathbf{x}$, we set $p(Y = y|\mathbf{x}) = 0$ where the constraint is violated and normalize the remaining density to 1. These distributions represent zero constraint violation (by definition) and can be obtained via rejection sampling from the baseline BNN or OC-BNN posteriors. Figure 3c shows the results of performing such rejection sampling, using the same setup as Figure 3b. As shown, rejection sampling is intractable for the baseline BNN (not a single SVGD particle accepted), but works well on the OC-BNN, even when using only a small sample count from $\mathcal{C}_\mathbf{x}$ to compute $p_\mathcal{C}(\mathbf{w})$. This experiment shows not only that naive rejection sampling (on ordinary BNNs) to weed out posterior constraint violation is futile, but also that *doing the same on OC-BNNs is a practical workaround to ensure that hard constraints are satisfied*, which is generally difficult to achieve in ideal, probabilistic settings.

## 6 Experiments with Real-World Data

To demonstrate the efficacy of OC-BNNs, we apply meaningful and interpretable output constraints on real-life datasets. As our method is the first such work for BNN classification, the baseline that we compare our results to is an ordinary BNN with the isotropic Gaussian prior. Experimental details for all three applications can be found in Appendix C.

### 6.1 Application: Clinical Action Prediction

| | Train | | Test | | |
|---|---|---|---|---|---|
| | Accuracy | $F_1$ Score | Accuracy | $F_1$ Score | Constraint Violation |
| **BNN** | 0.713 | 0.548 | 0.738 | 0.222 | 0.783 |
| **OC-BNN** | 0.735 | 0.565 | 0.706 | 0.290 | **0.136** |

Table 1: Compared to the baseline, the OC-BNN maintains equally high accuracy and $F_1$ score on both train and test sets. The violation fraction decreased about six-fold when using OC-BNNs.

The MIMIC-III database [12] contains physiological features of intensive care unit patients. We construct a dataset ($N = 405$K) of 8 relevant features and consider a binary classification task of whether clinical interventions for hypotension management — namely, vasopressors or IV fluids — should be taken for any patient. We specify two **physiologically feasible**, positive (deterministic) constraints: **(1)** if the patient has high `creatinine`, high `BUN` and low `urine`, then action should be taken ($\mathcal{C}_y = \{1\}$); **(2)** if the patient has high `lactate` and low `bicarbonate`, action should also be taken. The positive Dirichlet COCP (5) is used. In addition to **accuracy** and **$F_1$ score** on the test

|  |  | with race feature | | without race feature | |
|---|---|---|---|---|---|
|  |  | **BNN** | **OC-BNN** | **BNN** | **OC-BNN** |
| **Train** | Accuracy | 0.837 | 0.708 | 0.835 | 0.734 |
| | $F_1$ Score | 0.611 | 0.424 | 0.590 | 0.274 |
| | African American High-Risk Fraction | 0.355 | **0.335** | 0.309 | 0.203 |
| | Non-African American High-Risk Fraction | 0.108 | **0.306** | 0.123 | 0.156 |

Table 2: The OC-BNN predicts both racial groups with almost equal rates of high-risk recidivism, compared to a $3.5\times$ difference on the baseline. However, accuracy metrics decrease (expectedly).

set ($N = 69$K), we also measure $\epsilon$-satisfaction on the constraints as **violation fraction**, where we sample 5K points in $\mathcal{C}_{\mathbf{x}}$ and measure the fraction of those points violating either constraint.

**Results**   Table 1 summarizes the experimental results. The main takeaway is that **OC-BNNs maintain classification accuracy while reducing constraint violations**. The results show that OC-BNNs match standard BNNs on all predictive accuracy metrics, while satisfying the constraints to a far greater extent. This is because the constraints are intentionally specified in input regions out-of-distribution, and hence incorporating this knowledge augments what the OC-BNN learns from $D_{tr}$ alone. This experiment affirms the low-dimensional simulations in Section 5, showing that OC-BNNs are able to obey interpretable constraints without sacrificing predictive power.

## 6.2   Application: Recidivism Prediction

COMPAS is a proprietary model, used by the United States criminal justice system, that scores criminal defendants on their risk of recidivism. A study by ProPublica in 2016 found it to be racially biased against African American defendants [1, 16]. We use the same dataset as this study, containing 9 features on $N = 6172$ defendants related to their criminal history and demographic attributes. We consider the same binary classification task as in Slack et al. [24] — predicting whether a defendant is profiled by COMPAS as being high-risk. We specify the **fairness** constraint that the probability of predicting high-risk recidivism should not depend on race: for all ($\mathcal{C}_{\mathbf{x}} = \mathbb{R}^9$) individuals, the high-risk probability should be identical to their actual recidivism history ($\mathcal{D}_y$ is such that $p(y = 1) = $ `two_year_recid`). The AOCP is used. $D_{tr}$ is incompatible with this constraint since COMPAS itself demonstrates racial bias. We train on two versions of $D_{tr}$ — with/without the inclusion of race as an explicit feature. As the dataset is small and imbalanced, we directly evaluate the training set. To measure $\epsilon$-satisfaction, we report the **fraction of the sensitive attribute** (African American defendants vs. non-African American defendants) predicted as high-risk recidivists.

**Results**   Table 2 summarizes the results. By constraining recidivism prediction to the defendant's actual criminal history, **OC-BNNs strictly enforce a fairness constraint**. On both versions of $D_{tr}$, the baseline BNN predicts unequal risk for the two groups since the output labels (COMPAS decisions) are themselves biased. This inequality is more stark when the race feature is included, as the model learns the explicit, positive correlation between race and the output label. For both datasets, the fraction of the two groups being predicted as high-risk recidivists equalized after imposing the constraint using OC-BNNs. Unlike the previous example, OC-BNNs have lower predictive accuracy on $D_{tr}$ than standard BNNs. This is *expected* since the training dataset is biased, and enforcing racial fairness comes at the expense of correctly predicting biased labels.

## 6.3   Application: Credit Scoring Prediction

Young adults tend to be disadvantaged by credit scoring models as their lack of credit history results in them being poorly represented by data (see e.g. [14]). We consider the `Give Me Some Credit` dataset ($N = 133$K) [13], containing binary labels on whether individuals will experience impending financial distress, along with 10 features related to demographics and financial history. Motivated by Ustun et al. [26]'s work on *recourse* (defined as the extent that input features must be altered to change the model's outcome), we consider the feature `RevolvingUtilizationOfUnsecuredLines` (`RUUL`), which has a ground-truth positive correlation with financial distress. We analyze how much a young adult under 35 has to reduce `RUUL` to flip their prediction to negative in three cases: (i) a BNN trained on the full dataset, (ii) a BNN trained on a blind dataset (`age` $\geq 35$), (iii) an OC-BNN with an **actionability constraint**: for young adults, predict "no financial distress" even if `RUUL` is large. The positive Dirichlet COCP (5) is used. In addition to scoring accuracy and $F_1$ score on the entire test set ($N = 10$K); we measure the **effort of recourse** as the mean difference of `RUUL` between the two outcomes ($\hat{Y} = 0$ or 1) on the subset of individuals where `age` $< 35$ ($N = 1.5$K).

**Results** As can be seen in Table 3, the ground-truth positive correlation between `RUUL` and the output is weak, and the effort of recourse is consequentially low. However, the baseline BNN naturally learns a stronger correlation, resulting in a higher effort of recourse. This effect is amplified if the BNN is trained on a limited dataset *without* data on young adults. When an actionability constraint is enforced, the OC-BNN **reduces the effort of recourse without sacrificing predictive accuracy** on the test set, reaching the closest to the ground-truth recourse.

| | | Ground Truth | BNN (Full) | BNN (Blind) | OC-BNN |
|---|---|---|---|---|---|
| **Test** | Accuracy | | 0.890 | 0.871 | 0.895 |
| | $F_1$ Score | | 0.355 | 0.346 | 0.350 |
| | Effort of Recourse | 0.287 | 0.419 | 0.529 | **0.379** |

Table 3: All three models have comparable accuracy on the test set. However, the OC-BNN has the lowest recourse effort (closest to ground truth).

## 7 Discussion

**The usage of OC-BNNs depends on how we view constraints in relation to data.** The clinical action prediction and credit scoring tasks are cases where the constraint is a complementary source of information, being defined in input regions where $D_{tr}$ is sparse. The recidivism prediction task represents the paradigm where $D_{tr}$ is inconsistent with the constraint, which serves to correct an existing bias. Both approaches are fully consistent with the Bayesian framework, whereby coherent inference decides how the likelihood and prior effects each shape the resulting posterior.

In contrast with [7, 19, 25], **OC-BNNs take a sampling-based approach to bridge functional and parametric objectives.** The simplicity of this can be advantageous — output constraints are a common currency of knowledge easily specified by domain experts, in contrast to more technical forms such as stochastic process priors. While effective sampling is a prerequisite for accurate inference, we note that the sampling complexity of OC-BNNs is tractable even at the dimensionality we consider in Section 6. Note that we sample in $\mathcal{X}$-space, which is much smaller than $\mathcal{W}$-space.

**OC-BNNs are intuitive to formulate and work well in real-life settings.** Even though COCPs and AOCPs echo well-known notions of data-based regularization, it is not immediately clear that these ideas are effectual in practice, and lead to well-behaved posteriors with appropriate output variance (both within and without constrained regions). Our work represents the first such effort (i) to create a broad framework for reasoning with diverse forms of output constraints, and (ii) that solidly demonstrates its utility on a corresponding range of real-life applications.

## 8 Conclusion

We propose OC-BNNs, which allow us to incorporate interpretable and intuitive prior knowledge, in the form of output constraints, into BNNs. Through a series of low-dimensional simulations as well as real-world applications with realistic constraints, we show that OC-BNNs generally maintain the desirable properties of ordinary BNNs while satisfying specified constraints. OC-BNNs complement a nascent strand of research that aims to incorporate rich and informative functional beliefs into deep Bayesian models. Our work shows promise in various high-stakes domains, such as healthcare and criminal justice, where both uncertainty quantification and prior expert constraints are necessary for safe and desirable model behavior.

## Broader Impact

Our work incorporates task-specific domain knowledge, in the form of output constraints, into BNNs. We wish to highlight two key positive impacts. **(1)** OC-BNNs allow us to manipulate an **interpretable** form of knowledge. They can be useful even to domain experts without technical machine learning expertise, who can easily specify such constraints for model behavior. A tool like this can be used alongside experts in the real world, such as physicians or judges. **(2)** Bayesian models like BNNs and OC-BNNs are typically deployed in "high-stakes" domains, which include those with societal impact. We intentionally showcase **applications of high societal relevance**, such as recidivism prediction and credit scoring, where the ability to specify and satisfy constraints can lead to fairer and more ethical model behavior.

That being said, there are considerations and limitations. **(1)** If the model capacity is low (e.g. the BNN is small), constraints and model capacity may interact in unexpected ways that are not transparent to the domain expert. **(2)** Our sampling approach allows us to be very general in specifying constraints, but it also creates a trade-off between computational efficiency and accuracy of constraint enforcement. **(3)** Finally, the expert could mis-specify or even maliciously specify constraints. The first two considerations can be mitigated by careful optimization and robustness checks; the latter by making the constraints public and reviewable by others.

## Acknowledgments and Disclosure of Funding

The authors are grateful to Srivatsan Srinivasan, Anirudh Suresh, Jiayu Yao and Melanie F. Pradier for contributions to an initial version of this work [28], as well as Gerald Lim, M.B.B.S. for helpful discussions on the clinical action prediction task. HL acknowledges support from Google. WY and FDV acknowledge support from the Sloan Foundation.

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
