[Supplementary Material]

# A  Bayesian Inference over Neural Networks

On a supervised model parameterized by $\mathbf{W}$, we seek to infer the conditional distribution $\mathbf{W}|D_{tr}$, which can be computed from the data $D_{tr}$ using **Bayes' Rule**:

$$p(\mathbf{w}|D_{tr}) = \frac{p(\mathbf{w})p(D_{tr}|\mathbf{w})}{p(D_{tr})} \tag{11}$$

We call $p(\mathbf{w}|D_{tr})$ the **posterior** (distribution), $p(\mathbf{w})$ the **prior** (distribution) and $p(D_{tr}|\mathbf{w})$ the **likelihood** (distribution) [1]. Since $D_{tr}$ is i.i.d. by assumption, we can decompose the likelihood into individual observations:

$$p(D_{tr}|\mathbf{w}) = \prod_{i=1}^{N} p(Y_i|\mathbf{x}_i, \mathbf{w}) \tag{12}$$

The prior and likelihood are both modelling choices. The evidence probability $p(D_{tr})$ is an intractable integral

$$p(D_{tr}) = \int_{\mathcal{W}} p(D_{tr}|\mathbf{w}')p(\mathbf{w}')\,\mathrm{d}\mathbf{w}' \tag{13}$$

which is often ignored as a proportionality constant. We typically compute $p(\mathbf{w}|D_{tr})$ in log-form.

For a new point $\mathbf{x}'$, the predictive distribution over output $Y'$ is:

$$p(Y'|\mathbf{x}', D_{tr}) = \int_{\mathcal{W}} p(Y'|\mathbf{x}', \mathbf{w})p(\mathbf{w}|D_{tr})\,\mathrm{d}\mathbf{w} \tag{14}$$

where $p(Y'|\mathbf{x}', \mathbf{w})$ is the same likelihood as in (12). $Y'|\mathbf{x}', D_{tr}$ is known as the **posterior predictive** (distribution). Point estimates of $p(Y'|\mathbf{x}', D_{tr})$, e.g. the posterior predictive mean, can be used if a concrete output prediction is desired. Since (14) is intractable, we typically sample a finite set of parameters and compute a Monte Carlo estimator.

Performing Bayesian inference (11) on deep neural networks $\Phi_{\mathbf{W}}$ (with weights and biases parametrized by $\mathbf{W}$) results in a **Bayesian neural network** (BNN).

## A.1  Likelihoods for BNNs

The likelihood is purely a function of the model prediction $\Phi_{\mathbf{w}}(\mathbf{x})$ and the correct target $y$ and does not depend on $\mathbf{W}$ directly. As such, BNN likelihood distributions follow the standard choices used in other probabilistic models.

For regression, we model output noise as a zero-mean Gaussian: $\epsilon \sim \mathcal{N}(0, \sigma_\epsilon^2)$ where $\sigma_\epsilon^2$ is the variance of the noise, treated as a hyperparameter. The likelihood PDF is then simply the Gaussian $Y \sim \mathcal{N}(\Phi_{\mathbf{w}}(\mathbf{x}), \sigma_\epsilon^2)$.

For $K$-classification, we specify the neural network to have $K$ output nodes over which a softmax function is applied, hence the network outputs class probabilities [2]. The likelihood PDF is then simply the value of the node representing class $k$: $\Phi_{\mathbf{w}}(\mathbf{x})_k$.

## A.2  Priors for BNNs

For convenience and tractability, the common choice is an isotropic Gaussian $\mathbf{W} \sim \mathcal{N}(\mathbf{0}, \sigma_\omega^2 \mathbf{I})$, first proposed by MacKay [20], where $\sigma_\omega^2$ is the shared variance for all individual weights. Neal [21] shows that in the regression setting, the isotropic Gaussian prior for a BNN with a single hidden layer approaches a Gaussian process prior as the number of hidden units tends to infinity, so long as the chosen activation function is bounded. We will use this prior in the baseline BNN for our experiments.

## A.3  Posterior Inference on BNNs

As exact posterior inference via (11) is intractable, we instead rely on approximate inference algorithms, which can be broadly grouped into two classes based on their method of approximation.

**Markov chain Monte Carlo (MCMC)**   MCMC algorithms rely on sampling from a Markov chain whose equilibrium distribution is the posterior. In the context of BNNs, our Markov chain is a sequence of random parameters $\mathbf{W}^{(1)}, \mathbf{W}^{(2)}, \ldots$ defined over $\mathcal{W}$, which we construct by defining the transition kernel.

In this paper, we use **Hamiltonian Monte Carlo (HMC)**, an MCMC variant that employs Hamiltonian dynamics to generate proposals on top of a Metropolis-Hastings framework [5, 22]. HMC is often seen as the *gold standard* for approximate BNN inference, as (i) MCMC algorithms sample from the true posterior (which makes them more accurate than VI approximations, discussed below) and (ii) HMC is the canonical MCMC algorithm for BNN inference [3]. However, as HMC is inefficient and cannot scale with large or high-dimensional datasets, it is generally reserved for low-dimensional synthetic examples.

**Variational Inference (VI)**   Variational learning for NNs [8, 10] approximates the true posterior $p(\mathbf{w}|D_{tr})$ with a variational distribution $q_{\boldsymbol{\theta}}(\mathbf{w})$, which has the same support $\mathcal{W}$ and is parametrized by $\boldsymbol{\theta} \in \Theta$. The variational family $\mathbf{Q} = \{q_{\boldsymbol{\theta}}|\boldsymbol{\theta} \in \Theta\}$ is typically chosen to balance tractability and expressiveness. The Gaussian variational family is a common choice. To find the value of $\boldsymbol{\theta}$ such that $q_{\boldsymbol{\theta}}(\mathbf{w})$ is as similar as possible to $p(\mathbf{w}|D_{tr})$, we maximize a quantity known as the *Evidence Lower BOund* (ELBO):

$$\mathcal{L}_{\text{ELBO}}(\boldsymbol{\theta}) = \mathbb{E}_{\mathbf{W} \sim q_{\boldsymbol{\theta}}}\Big[ \log p(D_{tr}|\mathbf{W}) \Big] - D_{KL}\big(q_{\boldsymbol{\theta}}(\mathbf{w}) \,\|\, p(\mathbf{w})\big) \tag{15}$$

Estimators for the integral in (15) are necessary. Once a tractable proxy is formulated, standard optimization algorithms can be used.

In this paper, we use **Bayes by Backprop (BBB)**, a VI algorithm that makes use of the so-called *reparametrization trick* [15] to compute a Monte Carlo estimator of (15) on a Gaussian variational family [4]. BBB is scalable and fast, and therefore can be applied to high-dimensional and large datasets in real-life applications.

We also use **Stein Variational Gradient Descent (SVGD)**, a VI algorithm that relies on applying successive transforms to an initial set of particles, in a way that incrementally minimizes the KL divergence between the empirical distribution of the transformed particles and $p(\mathbf{w}|D_{tr})$ [17]. SVGD is also scalable to high-dimensional, large datasets. Furthermore, the transforms that SVGD apply implicitly define a richer variational family than Gaussian approximations.

## A.4   Prediction using BNNs

For all algorithms, prediction can be carried out by computing the Monte Carlo estimator of (14):

$$p(Y'|\mathbf{x}, D_{tr}) \approx \frac{1}{S} \sum_{i=1}^{S} p(Y'|\mathbf{x}', \mathbf{w}^{(i)}) \tag{16}$$

from $S$ samples of the (approximate) posterior. We can construct credible intervals for any given $\mathbf{x}'$ using the empirical quantiles of $\{\mathbf{w}^{(1)}, \ldots, \mathbf{w}^{(S)}\}$, which allows us to quantify how confident the BNN is at $\mathbf{x}'$.

# B   Approximations for BNN Prior Predictive

We state the approximate forms for a variational BNN prior predictive $p_{\boldsymbol{\lambda}}(\Phi_{\mathbf{W}}|\mathbf{x})$ in the case of regression and binary classification, where we assume a Gaussian variational family. Readers are referred to Chapter 5.7 of Bishop [3] for their derivations. We denote $\boldsymbol{\lambda} = (\boldsymbol{\mu}, \boldsymbol{\sigma})$ as the variational parameters for the mean and standard deviation of each $\mathbf{w}_i$.

**Approximation Prior Predictive for Regression**   A Gaussian approximation of the prior predictive is:

$$\Phi_{\mathbf{W}}|\mathbf{x} \sim \mathcal{N}(\Phi_{\boldsymbol{\mu}}(\mathbf{x}), \sigma_\epsilon^2 + \mathbf{g}^\top(\boldsymbol{\sigma}^2 \cdot \mathbf{g})) \tag{17}$$

where

$$\mathbf{g} = \left[\nabla_{\mathbf{w}}\Phi_{\mathbf{w}}(\mathbf{x})\Big|_{\mathbf{w}=\boldsymbol{\mu}}\right] \tag{18}$$

As noted in Appendix A, $\sigma_\epsilon$ is the standard deviation of the output noise that we model.

**Approximation Prior Predictive for Binary Classification**   We will need to make a slight modification to the BNN setup. Typically, a BNN for $K$-classification has $K$ output nodes, over which a softmax function is applied such that the output values sum to 1 (representing predicted class probabilities). Here, instead of using a BNN with 2 output nodes, we will use a BNN with a single output node, over which we apply the logistic sigmoid function [4]:

$$\sigma_L(x) = \frac{e^x}{e^x + 1} \tag{19}$$

The resulting value is then interpreted as the probability that the predicted output is 1 (the "positive" class):

$$\Phi_{\mathbf{w}}(\mathbf{x}) = p(Y = 1|\mathbf{x}) = \sigma_L(\phi_{\mathbf{w}}(\mathbf{x})) \tag{20}$$

where $\phi_{\mathbf{w}}(\mathbf{x})$ represents the output node's value *before* applying the sigmoid function. Note that $p(Y = 0|\mathbf{x}) = 1 - p(Y = 1|\mathbf{x})$. With this, the approximation for the prior predictive is given as:

$$p_{\boldsymbol{\lambda}}(\Phi_{\mathbf{W}}(\mathbf{x}) = 1|\mathbf{x}) = \sigma_L\left(\left(1 + \frac{\pi(\mathbf{g}^\top(\boldsymbol{\sigma}^2 \cdot \mathbf{g}))}{8}\right)^{-1/2} \mathbf{g}^\top \boldsymbol{\mu}\right) \tag{21}$$

where

$$\mathbf{g} = \left[\nabla_{\mathbf{w}}\phi_{\mathbf{w}}(\mathbf{x})\Big|_{\mathbf{w}=\boldsymbol{\mu}}\right] \tag{22}$$

Note that the first-order derivative $\mathbf{g}$ here is taken w.r.t. $\phi(\mathbf{x})$, not $\Phi(\mathbf{x})$.

# C   Experimental Details

All the details listed below can also be found at: \texttt{https://github.com/dtak/ocbnn-public}.

## C.1   Low-Dimensional Simulations

The model for all experiments is a BNN with a single 10-node RBF hidden layer. The baseline BNN uses an isotropic Gaussian prior with $\sigma_\omega = 1$. The output noise for regression experiments is modeled as $\sigma_\epsilon = 0.1$.

We run HMC for inference unless noted otherwise. When HMC is used, we discard 10000 samples as burn-in, before collecting 1000 samples at intervals of 10 (a total of 20000 Metropolis-Hastings iterations). $L = 50$, and $\epsilon$ is variably adjusted such that the overall acceptance rate is $\sim 0.9$. When SVGD is used, we run 1000 update iterations of 50 particles using AdaGrad [6] with an initial learning rate of 0.75. When BBB is used, we run 10000 epochs using AdaGrad with an initial learning rate of 0.1. $\theta = (\mu, \sigma)$ is initialized to 0 for all means and 1 for all variances. Each epoch, we draw 5 samples of $\epsilon$ and average the 5 resulting gradients. 1000 samples are collected for prediction.

In Figure 1, the hyperparameters for the negative exponential COCP are: $\gamma = 10000$, $\tau_0 = 15$ and $\tau_1 = 2$. In Figure 2, the hyperparameters for the positive Dirichlet COCP are: $\alpha_i = 10$ if $i \in \mathcal{C}_y(\mathbf{x})$, and 1.5 otherwise. In Figure 3a and Figure 4, we run 125 and 50 epochs of AOCP optimization respectively. $\lambda = (\mu, \sigma)$ is initialized to 0 for all means and 1 for all variances. The AdaGrad optimizer with an initial learning rate of 0.1 is used for optimization. In Figure 3b, the hyperparameters for the negative exponential COCP are: $\gamma = 10000$, $\tau_0 = 15$ and $\tau_1 = 2$. In Figure 6, the positive Gaussian COCP is used for all 3 constraints with $\sigma_{\mathcal{C}} = 1.25$. The training data is perturbed with Gaussian noise with mean 0 and standard deviation 1.

## C.2   High-Dimensional Applications

**Clinical Action Prediction**   The MIMIC-III database [12] is a freely accessible benchmark database for healthcare research, developed by the MIT Lab for Computational Physiology. It consists of de-identified health data associated with 53,423 distinct admissions to critical care units at the Beth Israel Deaconess Medical Center in Boston, Massachusetts, between 2001 and 2012. From the original MIMIC-III dataset, we performed mild data cleaning and selected useful features after consulting medical opinion. The final dataset contains 8 features and a binary target, listed below. Each data point represents an hourly timestamp, however, as this is treated as a time-independent prediction problem, the timestamps themselves are not used as features.

- `MAP`: Continuous. Mean arterial pressure. Standardized.
- `age`: Continuous. Age of patient. Standardized.
- `urine`: Continuous. Urine output. Log-transformed.
- `weight`: Continuous. Weight of patient. Standardized.
- `creatinine`: Continuous. Level of creatinine in the blood. Log-transformed.
- `lactate`: Continuous. Level of lactate in the blood. Log-transformed.
- `bicarbonate`: Continuous. Level of bicarbonate in the blood. Standardized.
- `BUN`: Continuous. Level of urea nitrogen in the blood. Log-transformed.
- `action` (**target**): Binary. 1 if the amount of either vasopressor or IV fluid given to the patient (at that particular time step) is more than 0, and 0 otherwise.

The model for all experiments is a BNN with two 150-node RBF hidden layers. For the baseline prior, $\sigma_\omega = 1$. BBB is used for inference with 20000 epochs. $\theta = (\mu, \sigma)$ is initialized to 0 for all means and 1 for all variances. The AdaGrad optimizer with an initial learning rate of 1.0 is used. As the dataset is imbalanced, the minority class (`action` $= 1$) is evenly upsampled. Points in $\mathcal{C}_\mathbf{x}$ were intentionally filtered from the training set. The training set is also batched during inference for efficiency. The positive Dirichlet COCP is used with $\alpha_0 = 2$ and $\alpha_1 = 40$.

**Recidivism Prediction**   The team behind the 2016 ProPublica study on COMPAS created a dataset containing information about 6172 defendants from Broward Couty, Florida. We followed the same data processing steps as [16]. The only additional step taken was standardization of all continuous features. The final dataset contains 9 features and a binary target:

- `age`: Continuous. Age of defendant.
- `two_year_recid`: Binary. 1 if the defendant recidivated within two years of the current charge.
- `priors_count`: Continuous. Number of prior charges the defendant had.
- `length_of_stay`: Continuous. The number of days the defendant stayed in jail for the current charge.
- `c_charge_degree_F`: Binary. 1 if the current charge is a felony.
- `c_charge_degree_M`: Binary. 1 if the current charge is a misdemeanor.
- `sex_Female`: Binary. 1 if the defendant is female.
- `sex_Male`: Binary. 1 if the defendant is male.
- `race`: Binary. 1 if the defendant is African American.
- `compas_high_risk` **(target)**: Binary. 1 if COMPAS predicted the defendant as having a high risk of recidivism, and 0 otherwise.

The model for all experiments is a BNN with two 100-node RBF hidden layers. For the baseline prior, $\sigma_\omega = 1$. SVGD is performed with 50 particles and 1000 iterations, using AdaGrad with an initial learning rate of 0.5. The dataset is batched during inference for efficiency. The AOCP is used with $\lambda = (\mu, \sigma)$ initialized to 0 for all means and 1 for all variances. 50 epochs of optimization are performed using AdaGrad at an initial learning rate of 0.1. We draw 30 samples from the convex hull of $D_{tr}$ each iteration to compute the approximation for (9). The optimized variance parameters are shrunk by a factor of 30 to 40 for posterior inference.

**Credit Scoring Prediction**    The `Give Me Some Credit` dataset ($N = 133$K) [13], taken from a 2011 Kaggle competition, contains 10 features and a binary target:

- `RevolvingUtilizationOfUnsecuredLines`: Continuous. Total balance on credit cards and personal lines of credit (except real estate and no installment debt like car loans), divided by the sum of credit limits.
- `age`: Discrete. Age of borrower in years. Standardized.
- `DebtRatio`: Continuous. Monthly debt payments, alimony, and living costs, divided by monthy gross income.
- `MonthlyIncome`: Continuous. Monthly income. Standardized.
- `NumberOfOpenCreditLinesAndLoans`: Discrete. Number of open loans and lines of credit. Standardized.
- `NumberRealEstateLoansOrLines`: Discrete. Number of mortgage and real estate loans, including home equity lines of credit.
- `NumberOfTime30-59DaysPastDueNotWorse`: Discrete. Number of times borrower has been 30 to 59 days past due (but no worse) in the last 2 years.
- `NumberOfTime60-89DaysPastDueNotWorse`: Discrete. Number of times borrower has been 60 to 89 days past due (but no worse) in the last 2 years.
- `NumberOfTimes90DaysLate`: Discrete. Number of times borrower has been 90 days or more past due in the last 2 years.
- `NumberOfDependents`: Discrete. Number of dependents in family, excluding themselves.
- `SeriousDlqin2yrs` **(target)**: Binary. 1 if the individual experiences serious financial distress within two years.

The model for all experiments is a BNN with two 50-node RBF hidden layers. For the baseline prior, $\sigma_\omega = 1$. BBB is used for inference with 10000 epochs. $\theta = (\mu, \sigma)$ is initialized to 0 for all means and $e$ for all variances. The AdaGrad optimizer with an initial learning rate of 0.1 is used. As the dataset is imbalanced, the minority class ($\text{action} = 1$) is evenly upsampled. The dataset is batched during inference for efficiency. The positive Dirichlet COCP is used with $\alpha_0 = 10$ and $\alpha_1 = 0.05$.

# D   Additional Results

Below, we show additional plots referenced in Section 5.

(a) Posterior predictive plot (SVGD).

(b) Posterior predictive plot (BBB).

Figure 5: Same experimental setup as in Figure 1, except that **(a)** uses SVGD and **(b)** uses BBB for posterior inference. Like HMC, both posterior predictive plots obey the constraints and fit $D_{tr}$. As in Figure 3b, SVGD produces an OC-BNN posterior that still violates the constraint to a small degree, as SVGD particles are optimized to be as far from each other (in $\mathcal{W}$) as possible. BBB produces a posterior that has smaller variance than HMC or SVGD everywhere in $\mathcal{X}$. Underestimation of variance is a commonly observed problem with VI methods.

Figure 6: The ground-truth function (dotted green curve) is $y^* = 5\cos(x/1.7)$. $D_{tr}$ contains 6 points perturbed by large Gaussian noise around ground-truth values. Three positive constraints (vertical green bands on $\mathcal{C}_\mathbf{x}$) are also placed using positive Gaussian COCPs around the corresponding ground-truth values. The resulting OC-BNN posterior predictive "overfits" to both the noisy data and all three constraints.

f

## Footnotes

[1] For probability distributions, we abbreviate $p(\mathbf{W} = \mathbf{w})$ as $p(\mathbf{w})$.

[2] A concrete label can be obtained by choosing the class with highest output value.

[3]MCMC methods "simpler" than HMC, such as naive Metropolis-Hastings or Gibbs sampling, are intractable on high-dimensional parametric models such as BNNs. As such, HMC is, in some ways, the simplest algorithm used for approximate BNN inference.

[4]The logistic sigmoid function derives its name from logistic regression, a simpler statistical model used for classification. The softmax function can be seen as a generalization of the logistic sigmoid function for $K > 2$.