[Reviews · NeurIPS 2020]

Review 1

Summary and Contributions: General framework for output constraints in BNNs, which is still novel and important contribution

Strengths: The experiments are in excellent

Weaknesses: The method description is technical and difficult to follow

Correctness: No issues

Clarity: Well written, but presentation is very technical

Relation to Prior Work: Yes

Reproducibility: Yes

Additional Feedback: Post-response update: The author response adressed my concerns very well, and the paper is good enough to be accepted, despite the lacking novelty. I am increasing my score to 7. ---- The paper proposes a new more general formalism to handle output constraints in BNNs. The space of constrained neural networks is already crowded, and while section 2 does make a good overview of the differences, it would greatly improve the paper to also define mathematically the differences in competing constraining methods and their scopes. Overall I had hard time understanding the contraint definitions (see below for minor comments). The presentation diverts from conventional “g=0” or “f \in C” constraints. The constraint formalism needs to be explicated better. Furthermore, at 4.1. I could not follow the math anymore. The authors need to either simplify their treatment; or if all the math is necessary, include detailed and reader-friendly explanations of the presentation in the appendix or in the main paper. The main idea still seems sound. The paper also doesn’t make it very clear how the huge space of w’s is handled, or how the estimation of the constraint densities is done in practise. Here, an algorithm box would have clarified the method greatly. The experiments are in general excellent, and demonstrate the methods behavior very well both in real life and simulated cases. The method works in general well with especially the fairness and multimodality examples being outstanding. However, there is also some limitations: in fig2b the green extrapolation looks odd, and in fig3a the constraints are still violated to some degree. There are no comparison any other constraint-based methods, which is a major drawback of the paper. Overall the paper presents a new formalism for BNN output constraints, and presents an excellent array of useful applications with great performance. The paper suffers from being difficult to read and grasp, and that the model/experiments are not reflected against competing methods. Minor comments: o def 4.1., I’m not sure if redefining the conventional (and simple) constraint form "g(f(x)) = 0” or “f(x) \in C” is necessary. I’m having hard time following this definition since it omits what “y” means, and what C_y(x) means (is it f(x), or a consraint-obeying f(x) or something else?). It seems that this definition says that for a certain “y”, all predictions with all x’s have to collapse to that single y. o def 4.2., the “y” is again undefined and not marginalized. I assume LHS should have “Y=y” instead of “Y”, otherwise one needs to marginalize over the assignments “y". Also it seems that only satisfied constraints would give positive values (due to the indicator), and then eps=0 should not refer to a hard constraint (but its opposite). o def 4.3., D_y has not been defined (does it relate to \Phi_w?). o The \eps’s of the two definitions should be different things, but the presentation implies that they are equal (since they share the symbol). For instance, discrepancies can be larger than 1. o eqs 8 and 9 do not contain \lambda, and q_lambda has not been defined. o fig1 has negative/exclusion constraints, but the figure shows positive/inclusion constraints. o The fig2 shows that the method works, but also leaks the green region away from the square in an arbitrary way. I would argue the baseline fit is more reasonable or even better. Please discuss.


Review 2

Summary and Contributions: The paper proposes to incorporate output constraints into BNN learning. It first gives a formal definition of the output constraints (deterministic and probabilistic), and then proposes two methods for inference. One is to use some soft data likelihoods that encourage the satisfaction of the constraints and then do posterior inference; the other is to directly optimize the prior parameters to obey the constraints to a maximum extent. The experiments show the usage of the proposed methods.

Strengths: 1. good motivation. Putting prior knowledge/constraint helps improves the interpretability of BNNs. Incorporating output constraints is more intuitive and easier. 2. The definition of the output constraints is formal and rigorous.

Weaknesses: 1. The methods are too straightforward and NOT novel enough. While the first approach is wrapped with many jargons from measure theory and stochastic process construction and seemingly deep, the key idea is to simply introduce some soft data likelihood to encourage the consistency to the constraint. The second idea is to directly optimize the definition. BTW, that is not called ``variational approximation'' because we do not see any variational representations. 2. Both inference methods are soft and can not guarantee the definition of the constraints can really be satisfied (i.e., \episilon level). These are in essence regularization methods, which are commonly strategies to incorporate existing knowledge. 3. I believe the definition 4.2 is wrong. You need to integrate out y as well. Otherwise, how is it consistent with Eq. (8)?

Correctness: mostly correct except the point mentioned above.

Clarity: yes

Relation to Prior Work: yes

Reproducibility: Yes

Additional Feedback: I have read the rebuttal. I do think the work is useful. I like the formalization of the output constraint better than the proposed algorithms (which are not novel enough to me). However, the gap between the constraint definition and how the algorithm satisfies the constraints should be addressed. Some theoretical guarantees about the proposed inference algorithm, e.g., the upper bound of the epsilon along with the number of data points, will make the work more solid and integral. I will maintain my score.


Review 3

Summary and Contributions: The paper outlines an approach to incorporating output constraints into Bayesian neural networks, forming the OC-BNN, with the ability to perform inference in function space rather than parameter space. The method is tested on several different real world datasets. ============ Post Rebuttal The two main discussion points were around clarity and novelty. Assuming the authors make the changes proposed, the clarity of the paper should be improved. The novelty is around producing a more general framework (than e.g. [18]) and a solid experimental setup. This is a solid (but not outstanding) contribution. My score remains the same (marginal accept) but I have increased my confidence in this score.

Strengths: - Tackles the important problem of more interpretable priors for BNNs - Empirical validation of the method - Entire framework has been developed

Weaknesses: - No discussion or experimental results varying epsilon (soft/hard constraints) - Not quite as novel as claimed

Correctness: In figure 2, the prior predictive sometimes falls outside of the constraint regions for the OC-BNN (whereas it doesn’t for the baseline methods) - why does this happen? Similarly, in figure 3, some of the posterior draws appear to violate the constraint region. Presumably this is because the constraint is a soft rather than hard constraint. Can you make it clear that this is the case, what the value of epsilon is, and show the comparison to hard constraints?

Clarity: The paper is mostly clear, although I found the derivation of the prior itself in §4.1 a bit unclear. For example, the authors say "(4) can loosely be viewed as an application of Bayes’ Rule, where the realization ΦCx of the stochastic process CP is the evidence (likelihood) of C being satisfied by w”. This looks more like a product of independent stochastic processes, so I don’t understand the link here. Also, is there an assumption that the constraints are independent, and don’t interact with each other?

Relation to Prior Work: When comparing to [18], the authors say that "Similar to OC-BNNs, constraint (C) satisfaction is modeled as the conditional p(Y, C|x), e.g. Gaussian or logistic. The main differences are: (i) their method is applied to deep Gaussian processes…”. Note that the DGP is in fact a stacked Bayesian linear model using random features, which is in fact *exactly* a BNN with a particular form of nonlinearity. Extending their approach to the classification setting would be relatively trivial as well (DGPs have been used in this setting, and the constraint formulation carries over). The point about amortisation holds, and the framework outlined is more general here, but the difference is not as clear as is being made out by the authors.

Reproducibility: Yes

Additional Feedback: Check capitalisation in references

[Author Response · NeurIPS 2020]

We thank all reviewers for their insightful comments. First, we will address some major themes from the reviews:

**Non-probabilistic Methods & Constrained Optimization**   Our paper is motivated by the ability to impose con-
straints probabilistically, i.e. the challenge is to incorporate constraints in a model that infers output *distributions*.
Bayesian techniques deviate significantly from classical optimization. As such, we did not include non-probabilistic
methods like [Stewart and Ermon, 2017] as baselines, or use notation conventional to constraint optimization.

**Hard vs. Soft Constraints**   Figure 3 shows minor violations of constraints as the priors used are soft and assign
small (but $> 0$) probability to infractions. The idiosyncratic nature of SVGD inference (used in Figure 3b) in learning
"repulsive/diverse functions" also makes it more likely to violate a constraint; for example, Figure S1 below shows that
for the same example, having only 10 SVGD particles eliminate the few violating functions.

More generally, in probabilistic systems, while hard constraints are theoretically
possible by assigning 0 probability to violations, (i) numerical instability issues
could arise, and (ii) we tend to obey Cromwell's rule in Bayesian inference, where
the support of the prior is usually the entire output space and unlikely functions are
naturally weeded out. **We stress that workarounds do exist**: (i) we can specify
extremely small ($\approx 0$) probability to the order of numerical insignificance, so
long as the prior remains differentiable, (ii) guarantees *on top of* soft constraints
can be enforced, e.g. rejection sampling over OC-BNNs (which will be tractable),
(iii) in the amortized setting, we can set a threshold for $\epsilon$ directly. We note that
soft constraints are often useful too, e.g. for learning (alongside the training data)
where the function might be *outside of* the constrained region.

Figure S1: Same example as Figure 3b, except using 10 SVGD particles (instead of 50).

**Novelty**   **(1)** While regularization techniques are common, it is not immediately
clear that data-based regularization leads to "well-behaved" and useful priors, (e.g.
smooth functions with suitable OOD variance), especially for the amortized variant, or for non-Gaussian likelihoods
(e.g. constraints over output ranges). Tractability of sampling with input dimensionality is also not obvious (and not
demonstrated by [18]), for example, we found that sampling at the border of constraints proved reasonably well at
guiding the model towards good functions. **(2)** We acknowledge points about similarity to [18] made by R4. A more
accurate comparison would be that our framework is more general and more versatile at incorporating a diverse range
of constraint formulations, without the need to make various Gaussian approximations or sacrifice tractability. **(3)** We
want to highlight the strength and novelty of our suite of experiments, which shows that OC-BNNs are useful and work
well on a diverse set of real-life problems and constraints.

**Additional Comments**

**[All]** *Technicality:* The stochastic process setup in Section 4 is to ensure a formal and principled definition keeping
with Bayesian inference, not to deceive the reader into unnecessary complexity. We acknowledge that a more intuitive
explanation and/or an explicit algorithm box might suffice and technical details be left to supplementary material.

**[R1][R3]** *Def. 4.2:* Indeed, we omitted marginalizing $y$. Equation (1) in Definition 4.2 should read:

$$p(Y \circ \mathcal{C}_y(\mathbf{x})|\mathbf{x}) = \int_{\mathcal{Y}} \mathbb{I}[y \circ \mathcal{C}_y(\mathbf{x})] \underbrace{\int_{\mathcal{W}} p(Y = y|\mathbf{x}, \mathbf{w})\, p(\mathbf{w})\, \mathrm{d}\mathbf{w}}_{\text{prior predictive}}\, \mathrm{d}y \quad \le \epsilon$$

Also, as R1 pointed out, $\circ$ should be swapped here: for a positive constraint, $\circ = \notin$ (and vice versa).

**[R3]** *Section 3:* We optimize w.r.t. the parameters of a Gaussian variational representation, hence "variational".

**[R1][R4]** *Fig. 2:* The imperfect fit is due to an idiosyncratic combination of low model capacity and sampling for this
particular example. Note that the plot was shaded at a specific confidence level; it is challenging for a 10-node RBF
network to fit a specific rectangle at *identical levels of confidence*. A key takeaway from Figure 2 is that we are not
overly confident far away from the green rectangle, especially in the prior predictive.

**[R4]** *Section 4.1:* There is a one-to-one correspondence: each constraint $(\mathcal{C}_{\mathbf{x}}, \mathcal{C}_y, \circ)$ is modeled with a *single* stochastic
process, whereby the points of the input region $\mathcal{C}_{\mathbf{x}}$ is the index of the process. (The equation on Line 135, on the other
hand, represents the product of multiple, independent constraints.) The confusion here may arise from the fact that the
points within a single constraint are also "independent" as their correct output has been directly defined by $\mathcal{C}_y$.

Russell Stewart and Stefano Ermon. Label-free Supervision of Neural Networks with Physics and Domain Knowledge.
In *Thirty-First AAAI Conference on Artificial Intelligence*, 2017.


[Meta-Review · NeurIPS 2020]

The reviewers highlighted the strengths of your experiments. While some questioned the novelty, I am definitely persuaded by the novelty and utility of your approach. I think you have made a very worthwhile contribution to the practical use of BNNs on many problems.